# Forward operator for polarimetric radio occultation measurements

Daisuke Hotta[1,2], Katrin Lonitz[3], and Sean Healy[1]

[1]European Center for Medium-Range Weather Forecasts, Shinfield Park, Reading, RG2 9AX, UK
[2]Meteorological Research Institute, Japan Meteorological Agency, 1-1 Nagamine, Tsukuba, Ibaraki, Japan
[3]European Center for Medium-Range Weather Forecasts, Robert-Schuman-Platz 3, 53175 Bonn, Germany

**Correspondence:** Daisuke Hotta (dhotta@mri-jma.go.jp)

**Abstract.** Global Navigation Satellite System (GNSS) Polarimetric Radio-Occultation (PRO) observations sense the presence of hydrometeor particles along the ray path by measuring the difference of excess phases in horizontally and vertically polarised carrier waves. As a first step towards using these observations in data assimilation and model diagnostics, a forward operator for GNSS-PRO observable $\Phi_{DP}$ (polarimetric differential phase shift) has been implemented by extending the existing two-dimensional forward operator for radio-occultation bending angle observations. Evaluation on heavy precipitation cases showed that the implemented forward operator can simulate very accurately the observed $\Phi_{DP}$ in synoptic-scale atmospheric river (AR) cases. For tropical cyclone cases it is more challenging to produce reasonable $\Phi_{DP}$ simulations, due to the highly sensitive of $\Phi_{DP}$ with respect to displacement of the position of the tropical cyclones. It was also found that snow is the dominant contributor to the simulated $\Phi_{DP}$, and that the ability to compute the ray paths in two dimensions is essential to accurately simulate $\Phi_{DP}$.

## 1 Introduction

The speed of light is slowed down when radio waves pass through the air, and this "retardation" is larger when the air is heavier and more humid. Because of this, as radio waves travel through stratified atmosphere from an emitter on a Global Navigation Satellite System (GNSS) satellite to a receiver onboard a lower-Earth orbit (LEO) satellite, they undergo bending (or refraction) to minimise the travel time along the ray. In Radio-Occultation (GNSS-RO) observations, this bending is retrieved from continuous measurement of the phase of the radio waves. As the refraction depends on the density of dry air and the amount of water vapour, measurements of bending can inform us about the thermodynamic properties of the atmosphere along the ray paths. GNSS-RO observations are routinely assimilated at most major NWP centres and are recognised as an indispensable component of modern NWP systems (e.g., Bonavita, 2014).

The carrier waves employed in GNSS are circularly polarised to minimise the impact of receivers' antenna alignment on the accuracy and stability of positioning. Because the carrier waves are polarised, it should be possible, in principle, to obtain information on properties of hydrometeors along the rays, just like polarimetric phase-shift measurement from dual-polarised weather radars (e.g., Kumjian, 2013). Polarimetric radar observations exploit the fact that, when polarised radio waves travel through a medium containing non-spherical objects like hydrometeor particles, the phase is delayed differently in the horizontally and vertically polarised waves due to the directionally differential cross-section of such objects, so that polarimetric

differential phase delay contains information about the presence of those particles along the ray. For example, if a large difference between horizontal and vertical phase shifts is observed, that indicates the presence of more and/or larger hydrometeors (and thus, heavier precipitation) along the ray since large rain droplets are usually oblate. This same principle should also hold for GNSS carrier waves to allow for inference of properties of hydrometeors from GNSS-RO observations if the horizontally and vertically polarised components of the radio waves can be processed separately, provided that depolarisation during the wave propagation through the ionosphere has been accounted for. An additional benefit of using GNSS carrier waves that are usually in L-band (India's regional navigation system Navic also used S-band in addition to L-band), in comparison to X-band to S-band as in most weather radars, is that the relatively low frequencies in the L-band range may make the measurement more sensitive to larger hydrometeors while being insensitive to smaller particles like aerosols and non-precipitating cloud droplets.

Such polarimetric measurement of GNSS-RO observations, which we shall call PRO hereafter, has not been explored until recently but was enabled by the sensor deployed for Radio Occultation and Heavy Precipitation (ROHP) mission onboard Spanish PAZ satellite (Cardellach et al., 2019). The PAZ satellite was successfully launched in May 2018 and has already been producing PRO measurements for more than 4 years (as of April 2023) with the observed cases including many heavy precipitation events.

PRO observation complements the standard RO measurements of a bending angle profile, which is sensitive to the thermodynamic variables (temperature, pressure and humidity) of the atmosphere, with additional information on the vertical profile of heavy precipitation. Such additional information is provided by measurement of the differential phase shift at each vertical level of the ascending/descending rays, which in turn is enabled by measuring the phase delay at two orthogonal (horizontal and vertical) polarisations.

The promise of PRO measurements is already established by recent studies. Cardellach et al. (2019) confirmed, with real data, that PRO measurements of differential phase shift exhibit stronger signals in the presence of heavier precipitation. Turk et al. (2021) and Padullés et al. (2021) simulated PRO measurements using hydrometeor retrieval products from collocated Global Precipitation Measurement (GPM) measurements and their results suggested that PRO measurements do contain useful information about vertical structure of deep convective clouds. Murphy et al. (2019) simulated PRO measurements of an airborne instrument using output data from an atmospheric meso-scale model and showed that such measurements can provide useful guidance on validation of microphysics representation in the model.

An important benefit that is unique to PRO observations is that, because the regular RO (or bending) measurement and the newly available PRO measurement are carried out simultaneously, profiles of thermodynamic and cloud-related properties can be observed at the same time. Hence, if an accurate observation operator is available that can simulate PRO measurements from state variables of a NWP model, PRO observations can potentially be of great diagnostic value to modelling of physical processes.

As a first step towards PRO assimilation and model validation with PRO measurements, we develop an off-line forward operator of PRO measurement for the European Centre for Medium-Range Weather Forecasts (ECMWF)'s Integrated Forecasting System (IFS) model.

The paper is structured as follows. Section 2 describes the specification and the main components of the forward operator, clearly presenting the assumptions we made and their potential limitations. Section 3 describes the data and the model used in this study along with the cases examined. Section 4 presents the results including those from several sensitivity experiments, followed by discussion and conclusions in section 5.

## 2    Description of the forward operator

The main observable of GNSS-PRO is the differential phase shift $\Phi_{\mathrm{DP}} = \Phi_H - \Phi_V$ which is the additional excess of the phase delay of the horizontal wave $\Phi_H$ in comparison to that of the vertical wave $\Phi_V$. This can be computed as the integration along the ray path, $s$, of the specific differential phase shift $K_{\mathrm{DP}}$:

$$\Phi_{\mathrm{DP}} = \int\limits_{\mathrm{GNSS}}^{\mathrm{LEO}} K_{\mathrm{DP}}(s)ds \tag{1}$$

where GNSS and LEO symbolically represent, respectively the position of the transmitter and receiver of GNSS radio signals.
$K_{\mathrm{DP}}$ indicates how much the phase of the horizontally polarised wave is delayed in comparison to that of the vertically polarised wave as they travel a unit distance. Since RO measurements represent path-integrated quantities, a positive value of $\Phi_{\mathrm{DP}}$ is an indication of the presence of horizontally-oriented hydrometeors somewhere along the ray path.

The main components to computing Eq. (1) are determination of the ray path, and estimation of $K_{\mathrm{DP}}$ from hydrometeors represented in the model. In IFS, hydrometeors from parametrised convection are represented as their vertical mass fluxes and
thus we need to convert convective mass fluxes to mixing ratio in order to relate them to $K_{\mathrm{DP}}$ (see section 2.3).

### 2.1    Ray-tracing

We develop the PRO forward operator by extending the operational two-dimensional (2D) forward operator for RO bending angle (Healy et al., 2007). The bending angle, $\alpha$, computed by this 2D forward operator, can be symbolically written as

$$\alpha = \int\limits_{\mathrm{GNSS}}^{\mathrm{LEO}} \left(\frac{d\alpha}{ds}\right) ds \tag{2}$$

with the ray path being identical to the one in Eq.(1). To compute the PRO observable $\Phi_{\mathrm{DP}}$, we exploit the analogy between Eq.(1) and Eq.(2) and use the existing code from the operational 2D bending angle operator to compute the ray path and to integrate the integrands.

The ray-tracing follows "Approach 2" of Healy et al. (2007) which is described in detail in their Section 3.2. For each $\Phi_{\mathrm{DP}}$ measurement, the latitude, longitude and the height of the tangent point, and the azimuth angle of the ray is taken from the
observed data. The forward operator then makes a 2D slice of the three-dimensional model field in the direction of the azimuth angle centred around the tangent point. The slice comprises vertical columns, each at model's native vertical full level, of equally-spaced locations along the occultation plane. Angular distance of two adjacent columns is set so that their horizontal

distance is approximately equal to the typical horizontal grid spacing of the input grid and the number of vertical columns in the slice is chosen so that the slice horizontally spans ∼1200 km. When the input model field is on 0.25°×0.25° regular lat-lon grid, for example, the forward operator first computes the latitudes and longitudes of 31 points equally spaced with $\Delta\theta = 40/6371$ radians (corresponding to 40 km physical distance at the Earth's surface) along the great circle centred around the tangent-point's horizontal position and with the azimuth angle specified by the observed data, and horizontally interpolates the model fields to these horizontal points to construct the 2D slice. Similarly, when the input model field is on 0.125°×0.125° regular lat-lon grid, the angular spacing is set as $\Delta\theta = 20/6371$ radians and the slice will contain 61 columns. In this study, for ease of implementation, the horizontal interpolation is done by nearest-grid search.

Once the 2D slice is set up, the forward operator computes the ray-path starting from the tangent point by integrating the ray equations in both directions (towards the receiver onboard LEO satellite and towards the transmitter onboard GNSS satellite). The numerical integration of the ray equations is based on the second-order Runge-Kutta method (Healy et al. (2007) used the fourth-order Runge-Kutta method but the operator was later simplified to adopt the second-order method). The $K_{\mathrm{DP}}$ contributions from each type of hydrometeors (see next section) are integrated as each section of the ray is traced, and the accumulated $K_{\mathrm{DP}}$ from each section is finally summed up to obtain the total $\Phi_{\mathrm{DP}}$. The detail of ray tracing is described in Healy et al. (2007). Vertical interpolation for $K_{\mathrm{DP}}$ is performed column-wise. While $K_{\mathrm{DP}}$ can be negative in nature in situations, for example, where particles in prolate orientation dominate over oblate ones, in our modeled formulation of $K_{\mathrm{DP}}$ (see the next subsection, in particular Eq. (3)), it always assumes non-negative values. It is hence desirable to ensure monotonicity in the interpolation, and for this reason we employ a simple linear interpolation in the vertical. Unlike refractivity, exponential decay with height is not assumed for $K_{\mathrm{DP}}$.

During an occultation event, the horizontal position of the tangent point drifts as the ray ascends or descends. While the operational 2D operator for bending angle accounts for such "tangent-point drift" (c.f., section 4.3) since 2011, Healy et al. (2007) found that assimilating regular RO observations of bending angle can be beneficial even when the effect of tangent-point drift is neglected. We found, from PAZ data, that in a single occultation event, the tangent point typically drifts ∼ 100 km. This can be tolerated for bending angle, but it is not clear if the same can be said for $K_{\mathrm{DP}}$ because it is sensitive to hydrometeors and their horizontal variability is much larger than that of thermodynamic fields. We examine this aspect in section 4.

We remark that the ray-tracing implemented in our 2D operator relies on the position of tangent points and the impact parameter provided from the RO data processing centres, but the tangent point position can only be determined after ray-tracing has been done. In this sense, our ray-tracing is dependent on externally performed estimation of the ray path. The accuracy of this tangent point estimation may impact the performance of our ray-tracing.

## 2.2 Relating hydrometeor water content to $K_{\mathrm{DP}}$

Specific differential phase shift, $K_{\mathrm{DP}}$, is induced by the difference in scattering properties of hydrometeor particles for the horizontally and vertically polarised waves. A "first-principle" scattering calculation of $K_{\mathrm{DP}}$ from NWP variables would require assumptions about details of cloud microphysics and precipitation that are not currently represented in the forecast

model. In IFS, the only prognostic variables related to hydrometeors are their water content (or vertical mass-flux; see next subsection), so $K_{\mathrm{DP}}$ needs to estimated from the water content variables from each type of hydrometeors.

In IFS, hydrometeors represented with the resolved (or large-scale) microphysics scheme can be categorised into the following four different kinds: non-precipitating liquid water, non-precipitating ice water, precipitating liquid water (or rain), and precipitating ice water (or snow). We denote the water content of these categories, respectively, by LWC, IWC, RWC and SWC. In addition to these resolved-scale variables, the deep convection parametrisation scheme also represents precipitating rain and snow separately. We denote the rain water content and snow water content that are attributable to the deep convection scheme, respectively, by RWCconv and SWCconv.

To determine $K_{\mathrm{DP}}$ from the water content of each category of hydrometeor, here we adopt a simple linear relation between the water content and $K_{\mathrm{DP}}$ contribution. For ice (*i.e.*, IWC, SWC and SWCconv), following Padullés et al. (2021), we adopt the following water-content-to-$K_{\mathrm{DP}}$ formula (which, in turn is taken from Bringi and Chandrasekar (2001)):

$$K_{\mathrm{DP}}(\mathrm{WC}) = \frac{1}{2}C\rho \times \mathrm{WC} \times (1 - ar) \tag{3}$$

where WC means any of IWC, SWC and SWCconv, in units of $\mathrm{g\,m^{-3}}$, $K_{\mathrm{DP}}(\mathrm{WC})$, in units of $\mathrm{mm\,km^{-1}}$, denotes the $K_{\mathrm{DP}}$ contribution from the hydrometeor whose water content is denoted WC, $\rho$ is the particle density in units of $\mathrm{g\,cm^{-3}}$, and $ar$ (non-dimensional) is the assumed axis ratio of the particles. $C$ is a proportionality constant which is derived from theory of scattering by spheroid objects which is set as $C = 1.6\,\left(\mathrm{g\,cm^{-3}}\right)^{-2}$ in this study. For $ar$ and $\rho$, we assume them to be constant and arbitrarily chose their values as $ar = 0.5$ and $\rho = 0.2\,\mathrm{g\,cm^{-3}}$.

The main approximating assumptions behind the formula relating $K_{\mathrm{DP}}$ to water content, Eq. (3), are (1) particles are spheroid, (2) particles are small enough in comparison to the wavelength so that Rayleigh scattering dominates, (3) $ar$ and $\rho$ are independent of the particle size (see the derivation in Chapter 7 of Bringi and Chandrasekar, 2001), and (4) the natural variability of the particle size distribution (PSD) can be ignored (*i.e.*, can be regarded fixed regardless of the atmospheric conditions). Compared with findings from weather radar community, the assumptions (1) and (2) may seem too crude for ice particles, but as we show later, the simulated $\Phi_{\mathrm{DP}}$ are quite consistent with the observations. We revisit this point in section 5.1. The assumption (3) is admittedly difficult to justify and we consider this to be an important limitation of our approach (see section 5.3).

It is relevant to note that $K_{\mathrm{DP}}$, at L-band in which Rayleigh limit is appropriate, is nearly proportional to the third moment of the PSD (Bringi and Chandrasekar, 2001, Section 7.1), which, in turn, is proportional to the water mass, unlike reflectivity which is related to the 6th moment of the PSD and thus is sensitive to the contributions from larger but fewer particles. Therefore, in comparison to radar reflectivity, $K_{\mathrm{DP}}$ is less sensitive to the precise shape of the PSD. In fact, as shown in Figure 11 in Turk et al. (2021), $K_{\mathrm{DP}}$ depends virtually linearly on the mass content. This figure also suggests that, if the particles are flat enough (i.e., if $ar$ is small enough), $K_{\mathrm{DP}}$ for ice particles can be larger than for rain particles. This, along with the fact that the rays of GNSS-(P)RO tend to travel longer in the upper atmosphere above the freezing levels than below, may explain the greater contributions to $\Phi_{\mathrm{DP}}$ from ice/snow particles than from liquid/rain particles (see discussion in Section 5.1).

We also note here that the orientation of the ice/snow particles are situation dependent and hence the axis ratio $ar$ would better be allowed to vary. For example, Padullés et al. (2021), when simulating $\Phi_{DP}$ using the ice water content retrieved from Global Precipitation Mission (GPM) Microwave Imager (GMI) observations, allowed $ar$ to vertically vary from 1 near the cloud top to 0 below $-10$ C°level and further modified it depending on the polarisation difference (PD) of the observed brightness temperature measured by GMI. Similarly, the particle density $\rho$ should be different depending on the particle size and shape. In our study, however, for simplicity, and due to lack of knowledge about particle orientation and details of particle shape, these effects are not accounted for.

The formula for liquid water (LWC, RWC and RWCconv) should be different from Eq. (3), but here, we use Eq. (3) for LWC, RWC and RWCconv as well. We are aware of the imperfection of this approach and aware that at least the coefficients should be refined. As we discuss in section 4, however, the $\Phi_{DP}$ contributions from liquid water are found not to be so dominant. Hence, small changes in tuning these parameters for liquid water will not change the main signals.

## 2.3 Converting mass flux to water content

In IFS, the amount of hydrometeor is represented (and archived) differently for the resolved (large-scale) scheme and for the parametrised convection scheme. In the resolved microphysics, LWC, IWC, RWC and SWC are directly available as specific water content (in units of $\mathrm{kg/kg}$), which can be readily converted to mass per volume (in units of $\mathrm{g\ m^{-3}}$). In the convective scheme, however, the amount of hyrdometeor (RWCconv and SWCconv) are represented only as their vertical mass fluxes (in units of $\mathrm{kg\ m^{-2}\ s^{-1}}$). To convert them into water content mass per volume, some additional assumptions have to be made. Here we follow the approach adopted in RTTOV-SCATT, described in Appendix B of Geer et al. (2007). In this approach, we assume that the particle density $\rho$ is constant, the fall speed of a particle of diameter $D$ is proportional to $D^{\beta}$ for some $\beta$ and the particle size distribution follows an exponential decay. With these assumptions, one can show (A. Geer, personal communication), with calculus involving Gamma function, that the water content WC, in units of $\mathrm{g\ m^{-3}}$, can be derived from the vertical mass flux FL as $\{\mathrm{FL}/(a\rho)\}^{1/b}$ with the parameters $a, b$ and $\rho$ given in Table 8 of Geer et al. (2007). This approach, adopted in RTTOV-SCATT, should be also applicable to models other than the IFS as long as vertical mass fluxes of water substances are available in their forecast or analysis product.

## 3 Model and data

### 3.1 Forecast model fields

The forecast data used to simulate $\Phi_{DP}$ are produced by running the version Cy47R3 of the IFS model at Tco1279 horizontal resolution (approximately 9 km of grid spacing) with 137 vertical model levels initialised from the operational analysis fields, available twice daily at 00 and 12 UTC, that are valid at the time closest to the start time of the occultation event simulated. Since the IFS Cy47R3 was operational from 12 October 2021 until 26 June 2023, the forecast fields used in this study are essentially identical to the operational deterministic forecasts (apart from technical differences such as use of different versions

of the compiler and some of the software stack) for the event that falls inside this period (tropical cyclone Kompasu, to be precise). For the other events that occurred before this period (c.f.,Table 1 and Table 2), the forecast fields used in this study are generated from a newer version of the IFS than the then-operational suite. The model fields defined on the model's native Tco1279 Octahedral Gaussian grid are interpolated to regular $0.125° \times 0.125°$ lat-lon grid by ECMWF's MARS system before being ingested to the forward operator.

The model fields are available at hourly interval in time. The forecast fields at two adjacent time steps are ingested to the forward operator, which are then linearly interpolated to the start time of the occultation event. The mass flux variables are archived as time accumulation from the beginning of the forecast integration; for these variables, we take the difference of forecast fields at two adjacent time steps that contain the time of occultation event, and divide the difference by 3600 s as if it is an instantaneous value, assuming that their values are constant over the one-hour time interval.

## 3.2 Observations

We use the version V06 of PAZ Level-1B data processed and calibrated at the Spanish Institute of Space Sciences ICE-CSIC/IEEC (Cardellach et al., 2019). The data are available for download from the ROHP-PAZ website (accessible at https://paz.ice.csic.es) upon registration. In this dataset, data for each occultation event are provided in a separate netCDF file containing time series of the calibrated $\Phi_{DP}$ and the tangent point height along with other relevant data and metadata that 200 includes the azimuth angle at the "occultation point" (*i.e.*, the point at which the excess phase first becomes greater than 500 m). To simulate $\Phi_{DP}$ with our forward operator we additionally need information on the latitude and longitude of the tangent point. These data are not included in the PAZ dataset, so we extracted them from the corresponding UCAR-processed level-2 data.

The $\Phi_{DP}$ data measured from PAZ is known to undergo height-dependent systematic errors and the PAZ dataset provides 205 $\Phi_{DP}$ data calibrated with two different approaches, one based on antenna pattern and the other based on linear regression. In this study we used the $\Phi_{DP}$ profile with antenna pattern calibration (denoted with the variable name `dphase_cal_ant` in the netCDF files) which was shown in Padullés et al. (2020) to be more accurate than the linear-regression-based calibration.

### 3.3 Examined cases

Using the forecast model and the observed data described above, $\Phi_{DP}$ profiles are simulated from the model and compared 210 with the observations for five atmospheric river (AR) cases and six tropical cyclone (TC) cases, which all exhibit large $\Phi_{DP}$ signals in the PAZ observations and are accompanied with heavy precipitation. The AR cases were selected by Dr. Michael Murphy, Prof. Jennifer Haase and Dr. Ramon Padullés, while the TC cases were selected by Dr. Ramon Padullés. These cases are proposed for a multi-center model intercomparison project of $\Phi_{DP}$ simulated from NWP output fields. Summary of the cases are given in Table 1 and Table 2.

| RO ID | time (UTC) | latitude | longitude |
|---|---|---|---|
| PAZ1.2020.355.18.18.G25 | 2020-12-20 18:17:52 | 51.36° N | 172.30° W |
| PAZ1.2020.356.05.00.G13 | 2020-12-21 04:59:51 | 44.66° N | 165.31° W |
| PAZ1.2021.014.15.53.G12 | 2021-01-14 15:52:44 | 45.60° N | 137.33° W |
| PAZ1.2021.009.02.43.G02 | 2021-01-09 02:43:09 | 48.83° N | 139.04° W |
| PAZ1.2021.010.03.58.G13 | 2021-01-10 03:58:06 | 38.69° N | 154.95° W |

**Table 1.** List of the examined Atmospheric River (AR) cases (provided by Dr. Ramon Padullés). RO ID is an identification code given following the UCAR convention for each occultation event; time (UTC) is the time at which the occultation begins; latitude and longitude are those of the "occultation point" where the excess phase exceeds 500 m for the first time during the occultation event.

| RO ID | time (UTC) | latitude | longitude | TC name |
|---|---|---|---|---|
| PAZ1.2018.143.03.04.G16 | 2018-05-23 03:03:48 | 14.50° N | 55.87° E | Mekunu |
| PAZ1.2019.303.09.35.G16 | 2019-10-30 09:35:10 | 14.37° N | 109.30° E | Matmo |
| PAZ1.2021.249.20.34.G29 | 2021-09-06 20:33:55 | 23.56° N | 54.64° W | Larry |
| PAZ1.2019.296.21.41.G14 | 2019-10-23 21:41:19 | 26.34° N | 141.57° E | Bualoi |
| PAZ1.2021.285.23.27.G04 | 2021-10-12 23:27:16 | 18.84° N | 112.65° E | Kompasu |
| PAZ1.2018.276.21.09.G25 | 2018-10-03 21:09:47 | 28.09° N | 57.61° W | Leslie |

**Table 2.** As in Table 1 but for Tropical Cyclone (TC) cases, with an additional column for the TC names.

## 4 Results

### 4.1 Baseline results

We first examine the overall agreement of the simulated and observed $\Phi_{DP}$ profiles, and the relative contributions from different categories of hydrometeors. The simulated and observed $\Phi_{DP}$ for AR and TC cases, plotted as vertical profiles against the tangent height, are shown in Figure 1. To use the best possible simulation as the baseline, in these plots the tangent-point drift is fully accounted for (*i.e.*, we use the correct tangent point position at each tangent height). To indicate the heights below which one can expect $\Phi_{DP}$ contributions from rain and liquid water, in each panel, the freezing level that is taken from the PAZ metadata is shown with a black thin horizontal line.

For the AR cases (Figure 1, top two rows), simulated $\Phi_{DP}$ profiles fit very well to the observed profiles albeit with some overestimation in two of the cases that occurred during January 2021. Considering all the uncertain assumptions that are made in linking hydrometeor water content to $K_{DP}$ (section 2.2), this level of agreement is quite surprising. From Figure 1 we can also observe that simulated $\Phi_{DP}$ is dominated by contributions from resolved-scale snow (SWC; yellow solid lines). Because of the uncertainty in how we estimate $K_{DP}$ from hydrometeor water content (section 2.2), we cannot assert that SWC contribution dominates solely by judging from their dominance in magnitude. However, the shape of the profile of SWC contribution closely resembles that of the observed $\Phi_{DP}$ profile for any of the five cases, which should mean that $\Phi_{DP}$ is predominantly determined

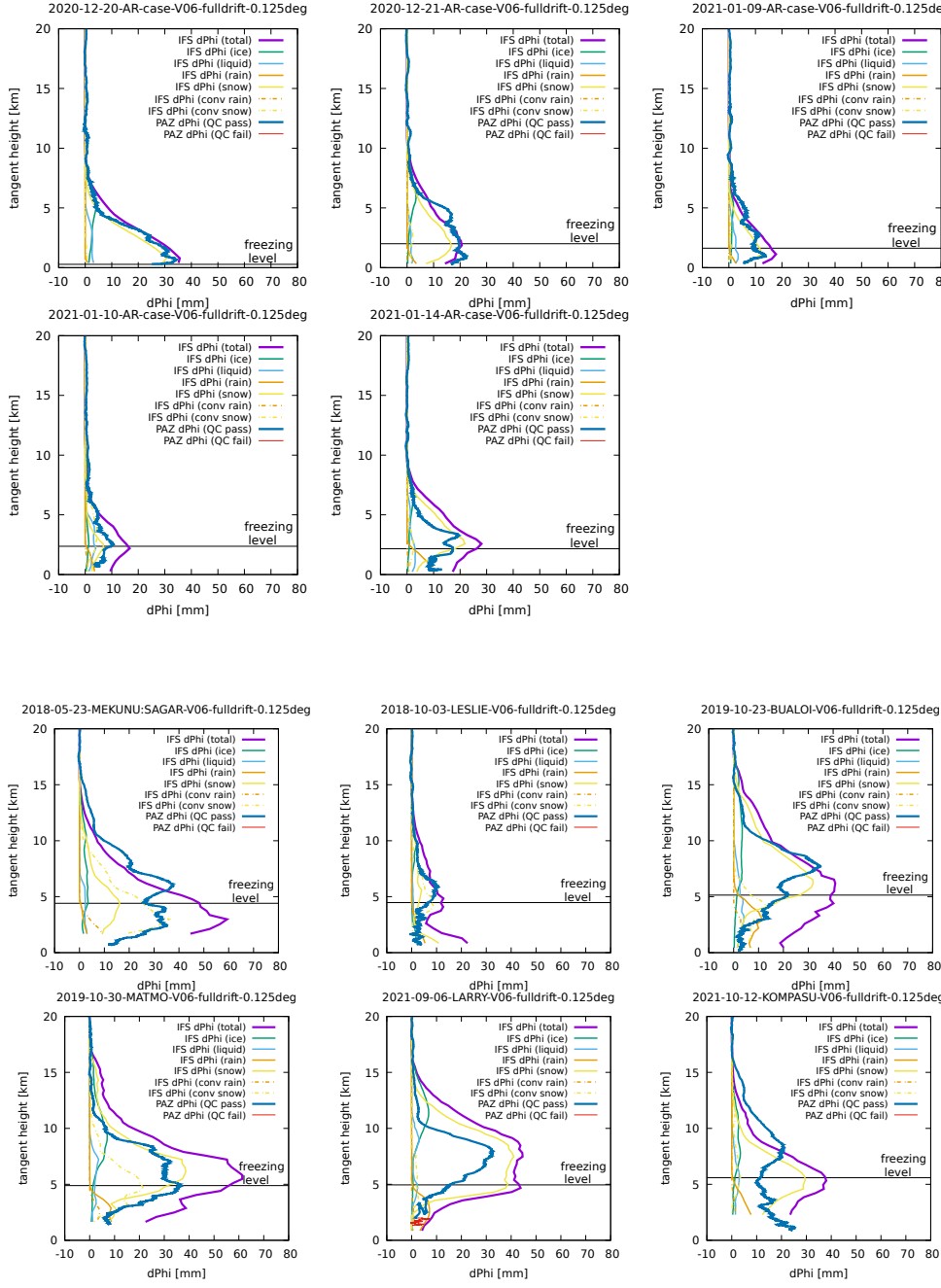

**Figure 1.** Comparison of the observed (blue) and simulated total (purple) $\Phi_{DP}$ profiles for the AR cases (top two rows) and the TC cases (bottom two rows). $\Phi_{DP}$ contributions from resolved-scale non-precipitating ice (IWC) and liquid (LWC), resolved-scale precipitating rain (RWC) and snow (SWC), and convective scheme rain (RWCconv) and snow (SWCconv) are also shown with different colours depicted in the legend. Black thin horizontal lines in each panel show the freezing level at the tangent point inferred from the retrieved temperature profile obtained from the PAZ RO data. Freezing levels roughly coincide with the levels at which RWC/LWC and RWCconv start to show non-zero contributions.

by resolved-scale snow. In contrast, for the TC cases (Figure 1, bottom two rows), the simulated and observed $\Phi_{DP}$ do not agree well, with the former significantly overestimating the latter. We investigate why the simulation results are so drastically different in AR and TC cases in the rest of this section.

## 4.2    Sensitivity to model-field displacement

We have seen that the simulated $\Phi_{DP}$ agrees well for AR cases but not so much for TC cases. One factor that may explain
this sharp contrast is the spatial scale of the phenomena: the horizontal scales of TCs are typically much smaller than those of synoptic disturbances like AR, so that even a small positional error in the model fields may have significant impact on simulated $\Phi_{DP}$ in the TC cases while these may be tolerated in the AR cases. Such a "displacement effect" was suggested earlier in Murphy et al. (2019).

     Estimates of the uncertainty in simulated $\Phi_{DP}$ that can be attributed to displacement of model fields are plotted in Figure
2. Here, we took a simple approach and estimated the uncertainty by shifting the latitude and longitude of the tangent points by $\pm 0.1^{\circ}$ (which correspond to shifts in position by $\sim 10$ km, namely, roughly by one grid point) or $0^{\circ}$, resulting in 9 profiles computed for each event in total. We assume that the spread among such profiles would represent the range of uncertainty that we would have if the forecast had displacement error on the order of one grid point. The IFS version Cy47R3, which we used to generate forecast fields assessed in this study, has tropical cyclone position error of $O(\sim 30)$ km on average in 0-12 hour
forecast range (Forbes et al., 2021, their Figure 6a). Thus, the displacement of $O(10)$ km that we examine here should yield reasonable estimates of $\Phi_{DP}$'s sensitivity to the background field uncertainties of particularly successful TC forecasts.

     Figure 2 shows that the simulated $\Phi_{DP}$ are insensitive to the forecast displacement in the AR cases but are more sensitive in the TC cases. This high sensitivity to the displacement error can explain the poorer fit between the simulated and observed $\Phi_{DP}$, albeit not the systematic overestimation in the TC cases.

## 4.3    Impact of tangent-point drift

The results shown so far have been computed by fully taking into account the effect of tangent-point drift (*i.e.*, by changing the horizontal position of the tangent point for each tangent-point height). In practice, this can be prohibitively expensive because, each time the tangent point position changes, the 2D slice has to be re-generated. It is thus desirable to reduce the frequency of tangent-point position update to minimise the number of 2D slices to be created as long as the accuracy is not too degraded.

Here, in addition to the "full drift" approach shown above, we explored two more approaches: "no drift", in which the drift of tangent point is not accounted for, and "11-batch", in which 11 neighbouring tangent point heights are grouped into a batch which shares the same 2D slice. In the 11-batch approach, rays in each batch are assumed to share the same tangent point horizontal position which is the 6th point of the 11 tangent points within the batch. The ECMWF's operational system uses the 11-batch approach to assimilate bending angles.

Profiles of $\Phi_{DP}$ simulated with the three approaches handling the tangent-point drift are shown in Figure 3. As we can expect from the insensitivity of simulated $\Phi_{DP}$ with respect to the horizontal displacement in AR cases, the different approaches yield $\Phi_{DP}$ that are equally consistent with the observations. In contrast, in TC cases, simulated $\Phi_{DP}$ profiles are highly dependent

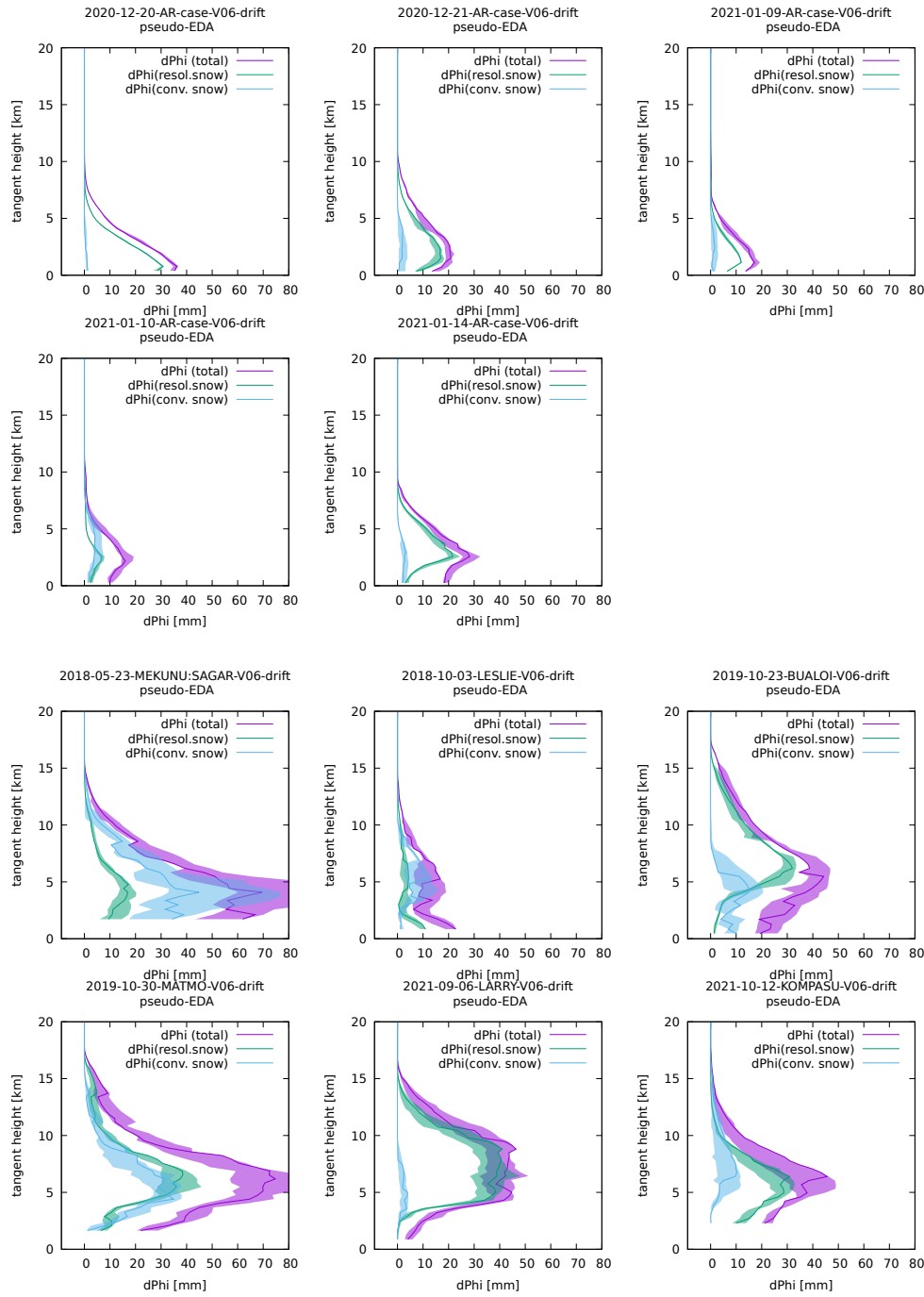

**Figure 2.** Uncertainty of simulated total $\Phi_{DP}$ (purple) and their contributions from resolved-scale snow (SWC; teal) and parametrised convective snow (SWCconv; pale blue), computed for each of the AR cases (top two rows) and each of the TC cases (bottom two rows). Here, the uncertainty is estimated by shifting or not shifting the latitude and longitude of the tangent points by $\pm 0.1°$. This results in computing $3 \times 3 = 9$ profiles in total, and the range between the minimum and maximum of such nine profiles are shown with shades. The unperturbed profiles are shown with solid lines.

on how the tangent-point drift is handled. Contrary to a naive expectation, however, the "full-drift" approach, which is the most expensive but should be the most accurate, does not necessarily result in simulations most consistent with the observations. This is likely because the overall error is dominated by the errors that result from displacement error and thus the impact from refining tangent-point drift is obscured.

## 4.4 Limitations of 1D operator

Most global NWP centres rely on a one-dimeinsional (1D) forward operator in simulating and assimilating RO bending angles. A 1D forward operator computes bending angle observable by only using the atmospheric profile at the tangent point, assuming that atmospheric thermodynamic fields within the occultation slice can be regarded horizontally uniform and hence an identical atmospheric profile can be used for all the columns in the 2D slice. It is thus interesting to see how a 1D operator would perform in simulating PRO observable $\Phi_{DP}$. Such an assessment will also clarify how essential 2D ray-tracing is in simulating realistic $\Phi_{DP}$.

Our 2D operator can easily operate in "1D mode" to emulate a 1D operator. To do this, we just set the derivative of the horizontal ray position to zero when integrating the ray equation (which is equivalent to assuming zero horizontal gradient of refractivity within the 2D slice). For simplicity, the tangent-point drift is ignored in our 1D computation.

The results of 1D computation are summarised in Figure 4. Unlike the 2D results (Figure 1), simulated $\Phi_{DP}$ are highly inconsistent with the observed $\Phi_{DP}$ even for the AR cases. The extreme case is the Hurricane Larry (centre panel in the bottom-most row) in which the simulated $\Phi_{DP}$ is almost zero except very near to the surface.

To understand why, cross-sections of resolved-scale snow water content (SWC), which is the dominant contributer to $K_{DP}$, are informative (Figure 5). In any of the cases, the distribution of SWC is far from being horizontally uniform, violating the assumption of the 1D computation. In the case of Hurricane Larry, the tangent point just happens to be inside the eye where there is no cloud and hydrometeors at all, so that 1D computation using only the profile at the tangent point completely misses out the hydrometeors in its vicinity.

The poor fit of the simulated and observed $\Phi_{DP}$ highlights the importance of the capacity to perform 2D computation in accurately simulating $\Phi_{DP}$ observable. The importance of 2D computation is not limited to PRO observations but also to regular RO observations of bending angle because the assumption of horizontal homogeneity can be questionable inside and in the vicinity of weather disturbances associated with heavy precipitation where the variability of thermodynamic parameters, especially humidity fields, tend to be large. This point will become increasingly important as the model's horizontal resolution gets even higher to better resolve precipitation systems.

## 5   Summary and discussion

A forward operator for GNSS-PRO observable $\Phi_{DP}$ has been implemented by extending the existing 2D forward operator for GNSS-RO bending angle observations assuming a linear relation between hydrometeor water content and $K_{DP}$. The implemented forward operator has been tested with five atmospheric river (AR) cases and six tropical cyclone (TC) cases which

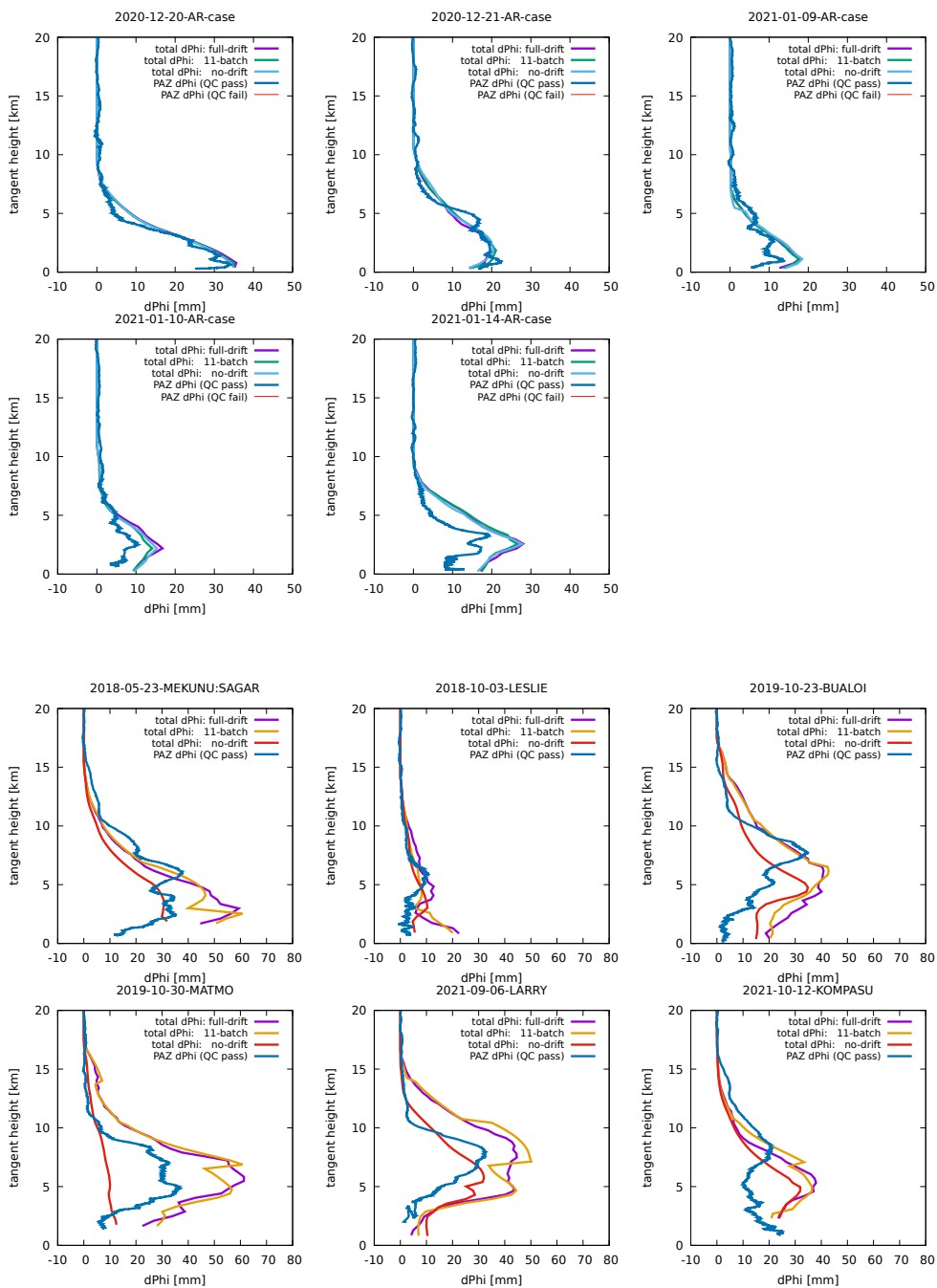

**Figure 3.** Impact of different approaches to account for tangent-point drifts. Profiles of $\Phi_{DP}$ simulated with the three different ways to handle the tangent-point drift (see text for detail) are plotted for each of the AR cases (top two rows) and each of the TC cases (bottom two rows), along with the observed $\Phi_{DP}$.

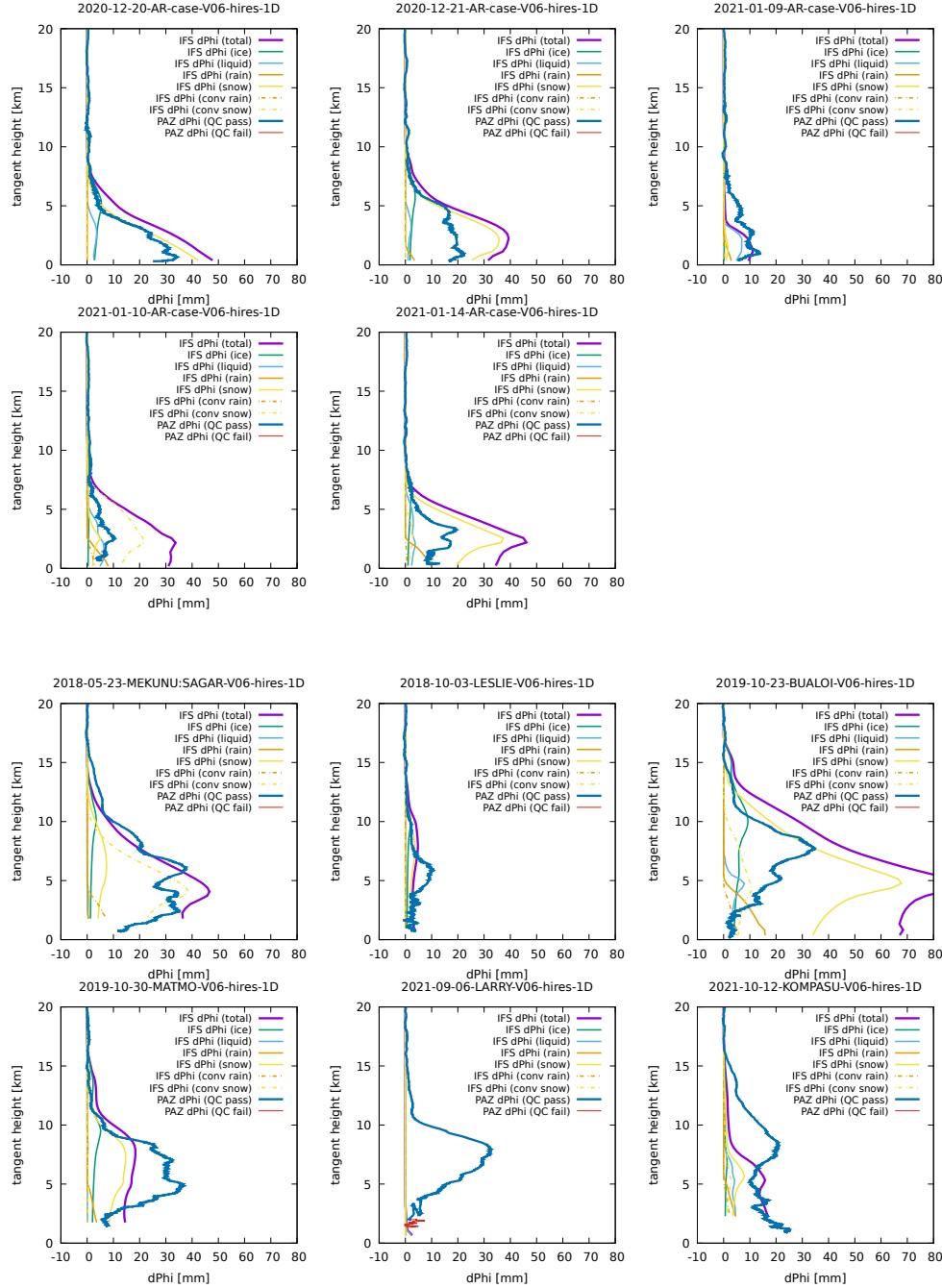

**Figure 4.** As in Figure 1, but with the profiles simulated with 1-dimensional computation.

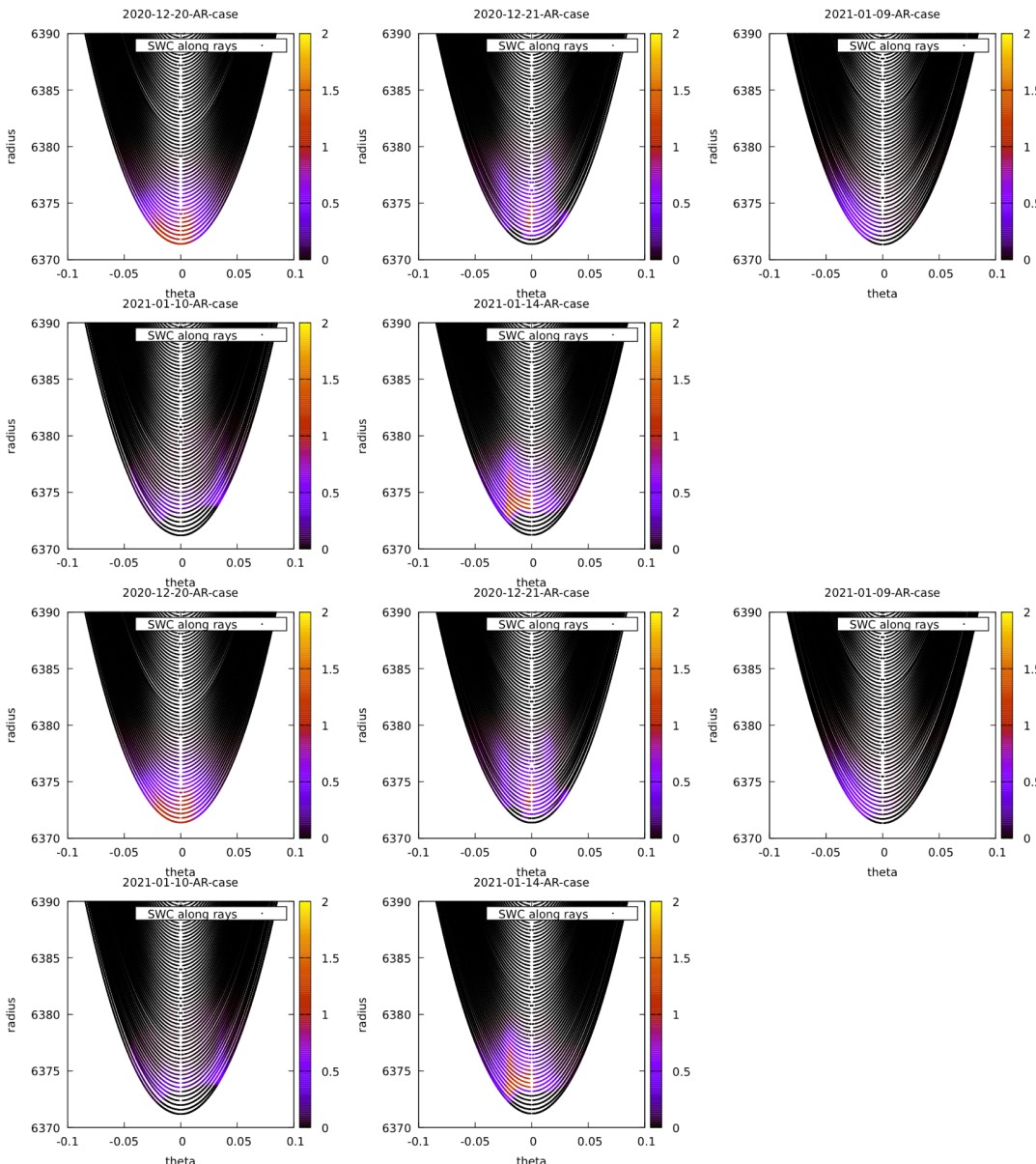

**Figure 5.** Resolved-scale snow water content in g m$^{-3}$ along the rays.

all accompanied heavy precipitation and strong $\Phi_{DP}$ signals in the actual observations. Despite all the simplifications, the implemented forward operator is found to be able to produce simulated $\Phi_{DP}$ profiles that are remarkably consistent with the corresponding observed profiles in most of the AR cases. Quantitatively speaking, however, the simulated $\Phi_{DP}$ profiles still tend to somewhat overestimate the observed profiles, and we speculate that such disagreement is likely attributable to imperfect assumptions on hydrometeor properties like the oblateness and the tilting angle (represented with the parameter $ar$), as we acknowledged in section 2.2, and the uncertainty of hydrometeor variables in the forecast fields. A more rigorous validation with more samples from different cases would be in order to draw a quantitative conclusion.

In contrast to the success with AR cases, we found TC cases to be much more challenging, with the simulated $\Phi_{DP}$ values systematically overestimating the observed $\Phi_{DP}$. Several additional $\Phi_{DP}$ simulations have been conducted with varying configurations of the implemented operator to understand its capacity and limitation. In this section we highlight and discuss the following main findings from this study.

## 5.1  Why snow dominates?

From the results in Figure 1 it was found that simulated $\Phi_{DP}$ is dominated by contribution from snow particles. This does not immediately mean that $\Phi_{DP}$ (or $K_{DP}$) is dominated by snow in reality because our modelled $K_{DP}$-water-content relation involves multiple uncertain assumptions and the hydrometeor representation in the IFS model is also subject to forecast uncertainties. Nevertheless, the remarkable agreement between the observed and simulated $\Phi_{DP}$ in AR cases suggests that the dominance of snow contributions to $\Phi_{DP}$ is likely realistic.

This finding is in line with previous findings by Turk et al. (2021) and Padullés et al. (2021) who simulated $\Phi_{DP}$ profiles observed by PAZ using liquid and solid water content retrieval from collocated cloud-sensitive measurements from other satellites. They found that liquid phase hydrometeor alone cannot explain the observed $\Phi_{DP}$ values, especially at high tangent point heights above freezing level. It appears, nevertheless, that liquid hydrometeor contributions to $\Phi_{DP}$ being negligible in comparison to snow has not been reported before.

In weather radar community, it is widely accepted that $K_{DP}$ per mass of snow particles is an order of magnitude smaller than that of liquid rain (Doviak and Zrnić, 1993) so that snow is largely undetectable from $K_{DP}$ measurements (e.g., Kumjian, 2013). Because of this, most forward operators that have been developed to simulate or assimilate radar $K_{DP}$ observations only consider warm rain conditions (e.g., Li and Mecikalski, 2013; Yokota et al., 2016; Kawabata et al., 2018a, b). From this perspective, it is surprising that PRO measurement likely senses predominantly the presence of snow rather than rain.

The geometry of GNSS-PRO may be a factor contributing to the high sensitivity of $\Phi_{DP}$ with respect to snow: the rays travel long distances at high altitudes except in the very vicinity of the tangent points as they go through the atmosphere from an emitter aboard a GNSS satellite toward the receiver aboard the PAZ satellite, so that, even if $K_{DP}$ values for snow/ice particles are relatively small compared to those for rain droplets (in fact, $K_{DP}$ for snow flakes that are flat enough can be larger than for rain droplets; see Figure 11 of Turk et al. (2021)), the integration of such smaller $K_{DP}$ over a much longer path than for rain droplets that are confined to lower altitudes below the freezing level can dominate in their contributions to the total $\Phi_{DP}$.

Another factor that may explain this apparent contradiction would be the difference in carrier wave frequencies of GNSS and weather radars. In GNSS, the L-band is chosen as the carrier frequencies since radio waves at these frequencies are less prone to attenuation by hydrometeors, thus allowing for signals to propagate in all sky condition. In the L-band, the frequency is $\sim 1.4$ GHz, corresponding to the wavelength of $\lambda \approx 20$ cm. In contrast, in weather radars, the carrier frequency of C- to X-band is typically chosen to maximise backscatter from hydrometeors, with a much shorter wavelength of $\lambda = 3 \sim 5$ cm. The longer wavelength of GNSS carrier waves makes the phase measurement more sensitive to the overall bulk properties of the hydrometeor particles than to their detailed shapes (Turk et al., 2021), whereby making $K_{\mathrm{DP}}$ at these frequencies more sensitive to snow particles than at lower frequencies, which may justify the simple linear $K_{\mathrm{DP}}$/WC relation.

## 5.2 Sensitivity to displacement

In section 4.2 we saw that the poor O-B fit for tropical cyclone (TC) cases is partly due to the high sensitivity of $\Phi_{\mathrm{DP}}$ to the displacement of clouds. Even a small shift in the latitude and longitude of only 0.1 degrees, which is equivalent to around 10 km (just one grid point of the deterministic high-resolution model), can lead to completely different simulations for TC cases. While this is helpful as it may inform the model about its incorrect TC positions through observations, it poses a methodological challenge for the data assimilation system.

Consider, for example, a scenario where the observed $\Phi_{\mathrm{DP}}$ is greater than the background $\Phi_{\mathrm{DP}}$ due to the misplaced TC position. In such a situation, positive O-B departures can be "corrected" in many different ways, such as locally increasing snow along the ray (which would be a wrong correction), changing the refractivity or temperature so that the ray passes through areas of intense snow (also a wrong correction), or shifting the position of the TC in the background fields (which would be the right correction). Out of these many possibilities, the data assimilation method needs to correct the background fields in the right way, but given the sparsity of GNSS-PRO observations, it is not obvious whether the information content provided by such observations is rich enough to constrain the correction in the right direction.

Apart from the sparsity of observations, correcting displacement of the background fields is also difficult because it is known to induce non-Gaussianity in the probability distribution of the background errors (e.g., Chen and Snyder, 2007; Aonashi et al., 2011).

## 5.3 Overestimation of $\Phi_{\mathrm{DP}}$ in TC cases

In this study, we have assumed a linear relation between $K_{\mathrm{DP}}$ and hydrometeor water content variables as we discussed in section 2.2. Despite such a simple assumption, our forward operator achieves remarkably good simulations for AR cases. Yet, the simulated $\Phi_{\mathrm{DP}}$ are systematically overestimated for TC cases, which deserves to be explored.

In our forward operator we assumed that the axis ratio $ar$ is constant at the arbitrarily chosen value of 0.5. While this choice resulted in $\Phi_{\mathrm{DP}}$ simulations that are in remarkably good agreement with the observations in AR cases, its validity may need to be reconsidered for TC cases. There are several observational evidence that $ar$ should be larger (*i.e.*, snow particles should be less horizontally oriented) in deep convective clouds than in stratiform clouds because strong turbulent mixing inside deep

convection randomises particle orientation (e.g., Gong and Wu, 2017). It may thus worth to allow $ar$ in our formulation to vary depending on the strength of mixing or vertical velocity of the background fields (Dr. Padullés, private communication).

## 5.4 Importance of 2D ray tracing

We found that, unlike the successful $\Phi_{\mathrm{DP}}$ simulations with the 2D operator, the 1D operator fails to reproduce the observed $\Phi_{\mathrm{DP}}$ even for AR cases where the 2D operator performs very well, which underlines the importance of incorporating the 2D
aspect in ray tracing calculation. This is in contrast to the case of regular GNSS-RO bending angle assimilation where a 1D operator is considered to be accurate enough to allow for extraction of meaningful information content from observations, although additional benefit is demonstrated with a 2D operator.

At the moment ECMWF is the only operational NWP centre to perform 2D ray tracing in assimilating GNSS-RO observations operationally. Our results suggest that, when other centres start investigation on GNSS-PRO assimilation, they would
need to start by first extending their RO forward operator to adopt 2D ray tracing. Depending on how the code is parallelised, this alone can be a non-trivial work.

## 5.5 Future directions

This study investigated characteristics of our implementation of a forward operator for GNSS-PRO observable $\Phi_{\mathrm{DP}}$. To the authors' knowledge, this is the first $\Phi_{\mathrm{DP}}$ forward operator ever implemented for an NWP model. While our first implementation
demonstrated promising results, especially for the synoptic-scale Atmospheric River (AR) cases, the results also identified several challenges that warrant further investigations.

The key challenge in assimilating PRO observations would be to account for displacement error of the background and this will be particularly important for smaller-scale phenomena such as tropical cyclones. While the currently operational 4DVar is known to be able to correct position errors in the background by assimilating dense observations like all-sky microwave
radiances (e.g. Duncan et al., 2022), it is not clear if such a correction is possible with horizontally sparse observations like GNSS-PRO and further methodological development along this line might be needed. We speculate that such correction will likely be possible if PRO observations are made available from a constellation of receiver satellites (e.g. Turk et al., 2019) to allow for dense sampling with multiple different azimuth angles.

The linear relation between $K_{\mathrm{DP}}$ and hydrometeor contents that we adopted is found to be quite successful despite its
simplicity, but its limitations have also been identified. To better account for a wider range of weather situations, it would be worth exploring a more elaborate $K_{\mathrm{DP}}$/WC relation. To this end, integration with RTTOV-SCATT would be beneficial because that allows the assumptions on microphysical properties like particle size distribution to be more consistent across different components of the NWP system.

In this study we focused on simulating the polarimetric differential phase shift $\Phi_{\mathrm{DP}}$ as the observable of GNSS-PRO,
but $\Phi_{\mathrm{DP}}$ is not the only GNSS-PRO observable. Wang et al. (2021) introduced polarimetric bending angle as an alternative observable quantity, and showed that polarimetric bending angle can be less prone to issues with multi-path, which may be

beneficial especially for measurements at low altitude. It would be thus worthwhile to also explore building forward operator for polarimetric bending angle.

*Code and data availability.* The PAZ data are available for download from https://paz.ice.csic.es/ upon registration. Some additional meta-data for PAZ data were retrieved from UCAR-processed level-2 data, which are publicly available from https://data.cosmic.ucar.edu/gnss-ro/paz/postProc/level2/ (doi: 10.5065/k9vg-t494). The forecast data were produced with ECMWF's IFS Cy47R3 suite. We made the IFS forecast data used in this study available for down load at doi:10.21957/hrkg-9c18. The forward operator for PRO observations developed in this study is based on the ROPP code which is available free of charge from https://www.romsaf.org/ropp/ after agreeing to license conditions and completing user registration. We plan to put the PRO code into the ROPP package in a later release, but the code is still under development. Every detail necessary to reproduce the PRO code is documented in section 2. To convert hydrometeor vertical mass fluxes to water content variables, we used code from RTTOV-SCATT which can be obtained at https://nwp-saf.eumetsat.int/site/software/rttov/ after agreeing to license conditions and completing user registration.

*Author contributions.* DH contributed to the conceptualisation, investigation, methodology, software, visualization and writing. KL contributed to the conceptualization, methodology, writing and software. SH contributed to conceptualisation, software, interpretation, and writing.

*Competing interests.* The authors declare no competing interests.

*Acknowledgements.* The authors thank Dr. Estel Cardellach and Dr. Ramon Padullés of ICE-CSIC/IEEC in Spain and Dr. Joe Turk of NASA/JPL for kindly providing PAZ data and for guiding us on the use and interpretation of their data. Dr. Michael Murphy and Prof. Jennier Haase are also acknowledged for selecting the AR cases examined. The authors also thank Peter Bechtold, Alan Geer, Robin Hogan and a number of other colleagues at ECMWF for their support. The authors thank Dr. Josep M. Aparicio and Dr. Joe Turk for their careful review of the manuscript and for constructive comments. This study is an outcome from a collaboration between ECMWF and Japan Meteorological Agency (JMA) financially supported by the Space-related Overseas Fellowship Program offered by the Ministry of Education, Culture, Sports, Science and Technology (MEXT) of the government of Japan.

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
