# Peer review of "Forward operator for polarimetric radio occultation measurements"

_Atmospheric Measurement Techniques, 2023_

## Referee Comment (RC2)

**Comments on "Forward operator for polarimetric radio occultation measurements "**

Numbers on left refer to line number in the online preprint:
https://amt.copernicus.org/preprints/amt-2023-132/amt-2023-132.pdf

This manuscript nicely describes early work towards development of a forward operator for the recently developed polarimetric RO measurement.   While the presentation is scientifically well presented, I have some minor comments below.

25.  The ionosphere also depolarizes the GNSS carrier.  Suggest to clarify, eg, "….if a large difference between horizontal and vertical phase shifts is observed (after accounting for ionospheric contributions), that indicates…."

30.  Also, India's Navic GNSS system indeed operates at S-band for improved positioning, but with no dual-polarization receive capability (that I am aware of).

Section 1, general.   To the less-informed reader, it's not clear what PRO provides "in addition to" standard RO.  Suggest making this clear in simpler terms.  "PRO complements the standard RO profile of bending angle with a complementary measure heavy precipitation at each vertical level.   This is done by measuring the phase delay at two orthogonal (horizontal and vertical) polarizations.  This additional measurement provides the path-integrated specific differential phase shift at each vertical level.  After inversion, a user therefore has in addition to the profile of temperature, moisture and pressure, a complementary measure of the presence of heavy precipitation along each ray path."   Or something like that… Right now, when you say, "…PRO measurements exhibit stronger signals in the presence of heavier precipitation" it is not clear what signal or measure you are referring to.

62.  Slight correction:  "Since RO measurements represent path-integrated quantities, a positive value of $K_{DP}$ is an indication of the presence of horizontally-oriented hydrometeors somewhere along the ray path".

125.  The uncertainty in (3) also relates to the natural variability in the unknown particle size distribution (PSD).   Near L-band, Kdp is more related to the mass water content ($3^{rd}$ moment of the PSD), unlike radar reflectivity ($6^{th}$ moment of the PSD).  Therefore, Kdp is less sensitive to variations in the PSD and the contributions from the largest drops.

For "flat enough" ice particles, Kdp can actually larger than for (similar water content of) rain media.   Since the heavy liquid phase only rain is usually concentrated in a smaller domain than the rest of the (mixed or solid) phase above, a small Kdp integrated over a long path can exceed that from a larger Kdp but integrated over a shorter path.    Suggest you cite Fig 11 of Turk et al 2021 to give the reader an idea of the range of values being talked about (pasted below).

[Figure]

FIG. 11.

One-way propagation differential phase shift $K_{dp}$ (° km$^{-1}$, left axis) at 1.4 GHz, as a function of equivalent mass content $M$ (g m$^{-3}$). The blue, green, and red lines indicate an ice media characterized by oblate spheroids with axis ratios of 0.8, 0.5, and 0.2, respectively, and black is for a rain media with axis ratio following Beard and Chuang (1987). The associated colored dashed lines indicate the propagation distance (km, right axis) through each of these media that would be required for a value of $\Delta\phi = 10$ mm.

Citation: Journal of Atmospheric and Oceanic Technology 38, 10; 10.1175/JTECH-D-21-0044.1

⊕ Download Figure

⊕ Download figure as PowerPoint slide

⬚ View Full Size

138.  Presumably the drop size in the IFS representation of LWC are so small, that these are essentially spherical anyhow (and hence contribute little to no to the net $\phi_{DP}$ ).

150.  Is the Geer et at formulation for converting vertical mass flux to water content specific to the IFS model, or is this a more general formulation?  Readers of this manuscript may want to use this formulation with other models or reanalysis that provide vertical mass flux variables.

156.  When you say, "…This configuration is essentially identical to the operational deterministic forecast", is this implying that since the simulation is done with model data so close in time to one of the operational model output time steps, that it is "essentially identical"?

200.  Suggest that you cite the work of Murphy et al who earlier suggested this "displacement effect" in their 2019 paper (see their discussion on observation geometry).  In fact, you should cite this paper in the very beginning of this manuscript, as their work is the first study on microphysics and PRO.

[1]
M. J. Murphy, J. S. Haase, R. Padullés, S.-H. Chen, and M. A. Morris, "The Potential for Discriminating Microphysical Processes in Numerical Weather Forecasts Using Airborne Polarimetric Radio Occultations," *Remote Sensing*, vol. 11, no. 19, Art. no. 19, Jan. 2019, doi: 10.3390/rs11192268.

230.   I would suggest renaming this section to, "Limitations of 1D Operator".  This is an important section and these limitations will magnify as model resolutions further shrink.

It's important to mention that this limitation is not only valid for modeling PRO in precipitation (as you show), but in general for modeling of RO, e.g., the bending angle from the water vapor field.   In areas of high pressure and fair weather (no clouds), one would expect little different between the bending angle simulated by the 1D and 2D operators as the horizontal water vapor field is rather homogeneous.   But near clouds and precipitation, where one wants to maximize the benefit of PRO or RO observations, there are also variations in the water vapor field (eg, dry/moist across frontal boundaries) such that simulated bending angles can be quite different depending upon whether the forward operator was positioned "across" or "along" the weather front.

With a coarse model grid spacing (eg, 1-degree or so), these effects may not matter too much (?). But as model resolutions get smaller and smaller, and resolve clouds and water vapor at km

scales, it is important to replicate the actual ray path viewing geometry to being simulations and observations into accord.

325.  Referring to "horizontally sparse observations", if a dense sampling of PRO-like observations were to be available (each PRO viewing from a different relative azimuth angel), do you think that PRO would also be able to better correct for position errors?

---

## Author Response (AR1)

**Response to reviewer #1**

Reply to the review by reviewer # 1 (Dr. Josep M. Aparicio) of **AMT-2023-132: "Forward operator for polarimetric radio occultation measurements"** by Daisuke Hotta, Katrin Lonitz, and Sean Healy

We would like to express our sincerest gratitude for spending your precious time on kindly reviewing our manuscript. We believe that revision of the manuscript following your thoughtful suggestions have certainly allowed us to significantly improve the manuscript, especially in terms of precision and clarity.

Please find below our point-by-point responses to you comments. Your comments are cited in red colour, followed by our replies in black. For references cited in the responses below, please see the list of references provided in the revised manuscript.

For convenience, we have attached a tracked-changes version of the revised manuscript (`diff.pdf`) generated with the `latexdiff` utility; in this file, newly added and deleted texts are highlighted, respectively, with blue and red colours. Line numbers and page numbers shown in the responses below refer to those in the `diff.pdf` file.

We hope that our revision succeeded in addressing all your concerns, and thank you again for your careful assessment.

**Point-by-point responses**

The manuscript presents an interesting exploration of the ability of an NWP system to estimate observed RO polarization observations, a precondition to assimilate them. The authors find qualitatively good forward estimates, with an ability to identify the effects of rain and snow. Although the agreement is still not accurate in the quantitative sense, this can be interpreted as both the uncertainty of the detailed hydrometeor field, and a margin associated with the oblateness, and probably the tilt, of hydrometeors, quantities with large uncertainty.

L23-24: "radio waves travel through oblate objects"

The principle is true both with oblate and prolate objects (hydrometeors may be both). Also, a large fraction of the effect happens because radio waves travel next to such objects (not just through). I suggest a minor adaptation of the sentence, for instance "radio waves travel through a medium containing ellipsoidal objects".

*Response:*

Thank you for pointing out our imprecision. We agree that $K_{DP}$ can be negative if the wave passes through prolate objects. We corrected the description as suggested to make it more precise (L24-29).

L92: "to avoid negative values": Would it be inappropriate to have negative values? I understand that it may be desirable to maintain the monotonicity of the interpolated field, and not contaminate it with a spurious wavy interpolation, but please correct me if I am wrong (or better, specify in the paper): is the differential phase shift necessarily positive? I would guess that it could be positive or negative, as a function of the oblateness/prolateness of hydrometeors, and one could have both along a line of sight. At least, I understand that atmospheric snow/ice can appear prolate in polarimetric radar. Please elaborate/clarify.

*Response:*

We thank you again for pointing this out. It is true that $K_{DP}$ in nature can be negative depending on the orientation of the particles through which the wave passes. Here in this context, however, we are discussing interpolation of the $K_{DP}$ field that is simulated from model forecasts,

and in our formulation described in Section 2.2, $K_{\mathrm{DP}}$ is always non-negative, so that monotonicity implies avoidance of negative values. We made this point clear in the revised text (L103-L106).

L95: "tangent point drift is not crucial for the regular RO observations". It looks oddly expressed. It was found to be important, albeit it is indeed a small fraction. I agree that the difference must be much larger for hydrometeors.

*Response:*

We agree with the reviewer that "not essential" was not the right word to describe the role of accounting for tangent point drift. Following your suggestion, we modified the text in the revised manuscript (L111-112) removing the phrase "the effect of tangent-point drift is not crucial" and wrote that *(assimilation of bending angle) can be beneficial even when the effect of tangent-point drift is neglected.*

Also, L95 "presumably due to the weak horizontal gradient". The word "presumably" also looks odd, as it is quite established that for this very fact one gets a perceivably less accurate, although still reasonable, result if the drift is ignored. Consider rephrasing.

*Response:*

Again we agree with the reviewer and removed the specified phrase (L111).

L122: "axis ratio of the ice particles" Since this is finally applied to water also, should it not "axis ratio of the particles", thus including water droplets? Also, it looks strange that the water shape vs rain rate relationship being relatively well established, it turns out to be less developed in the paper.

*Response:*

We agree and dropped "ice" preceding "particles" in the text below Eq. (3) that explains notation used there (L138).

It is true that our treatment of rain and liquid particles is too simplistic. Refining liquid water treatment is beyond the scope of our current paper but we do plan to explore a more rigorous approach, as we mentioned in the section on "future directions" (L381-L385). As we stated there, we plan to incorporate formulations adopted in RTTOV-SCATT to make our operator more consistent with the other components of the NWP system, and this includes more sophisticated simulation of $K_{\mathrm{DP}}$ for rain that leverages more advanced understanding of scattering by rain and liquid particles.

Given the importance of the ice vs liquid water phase that is being featured in the manuscript, could some approximate indicator of the height of the freezing point be added to some figures (notably the panels in Fig 1). To some extent, it is visible by the level where the rain signal becomes non-zero in each panel (in the range 2-6 km). If that curve happened indeed to be a reasonable indicator, you may mention it in the caption.

*Response:*

We thank the reviewer for this useful suggestion. We also had the same suggestion from Prof. Jennier Haase in her community comment (https://doi.org/10.5194/amt-2023-132-CC1). Following the suggestion, we replotted Figure 1 with the freezing levels indicated with black thin horizontal lines. Consistent with your expectation, these levels do roughly coincide with the levels at which rain/liquid begin to show non-zero contributions. We revised the text accordingly

(L223-225 and the caption of Figure 1).

Sec 4.2: Sensitivity to displacement. A 10 km displacement is introduced. It is however not mentioned whether this is a good/inaccurate estimation of the geographic accuracy of IFS. Presumably, it was selected because this is order of magnitude the position error of AR and TC features within IFS at short lead time (say, about 6h). Is that so? Please comment whether these figures are indeed representative of IFS's accuracy.

*Response:*

We thank the reviewer again for raising this point. From tropical cyclone forecast verification, we know that in IFS Cy47R3, the average tropical cyclone position error for 12 hour forecasts, verified against best track analysis, is about 30 km. Considering the error of best track analysis itself, and taking into account that the first-guess used in data assimilation is usually a shorter-range forecast (on average a 6-hour forecast for a 12-hour assimilation window), we consider that the $\sim$ 10 km displacements that we examined in this study are commensurate with cases of successful TC forecasts. Unfortunately we do not have quantitative estimate of position errors for AR cases, but we speculate that the forecast accuracy for TC and AR cases is not too different. In the revised text, we provided some explanation about these (L245-248).

**Response to reviewer #2**

Reply to the review by reviewer # 2 (Dr. Joe Turk) of **AMT-2023-132: "Forward operator for polarimetric radio occultation measurements"** by Daisuke Hotta, Katrin Lonitz, and Sean Healy

We would like to express our sincerest gratitude for spending your precious time on kindly reviewing our manuscript. Your thorough review and a number of insightful suggestions have certainly allowed us to significantly improve the manuscript.

Please find below our point-by-point responses to you comments. Your comments are cited in red colour, followed by our replies in black. For references cited in the responses below, please see the list of references provided in the revised manuscript.

For convenience, we have attached a tracked-changes version of the revised manuscript (`diff.pdf`) generated with the `latexdiff` utility; in this file, newly added and deleted texts are highlighted, respectively, with blue and red colours. Line numbers and page numbers shown in the responses below refer to those in the `diff.pdf` file.

We hope that our revision succeeded in addressing all your concerns, and thank you again for your careful assessment.

**Point-by-point responses**

This manuscript nicely describes early work towards development of a forward operator for the recently developed polarimetric RO measurement. While the presentation is scientifically well presented, I have some minor comments below.

25. The ionosphere also depolarizes the GNSS carrier. Suggest to clarify, eg, "....if a large difference between horizontal and vertical phase shifts is observed (after accounting for ionospheric contributions), that indicates...."

*Response:*

Thank you for pointing this out. We revised the text to mention the need for correcting ionospheric effect (L31-33). Note that, because the sentence you suggested to modify is made in the context of ground-based weather radar for which ionospheric effect is not present, we instead modified the text which introduces PRO for the first time.

30. Also, India's Navic GNSS system indeed operates at S-band for improved positioning, but with no dual-polarization receive capability (that I am aware of).

*Response:*

Thank you for the information. We added this information in the footnote 1 in page 2.

Section 1, general. To the less-informed reader, it's not clear what PRO provides "in addition to" standard RO. Suggest making this clear in simpler terms. "PRO complements the standard RO profile of bending angle with a complementary measure heavy precipitation at each vertical level. This is done by measuring the phase delay at two orthogonal (horizontal and vertical) polarizations. This additional measurement provides the path-integrated specific differential phase shift at each vertical level. After inversion, a user therefore has in addition to the profile of temperature, moisture and pressure, a complementary measure of the presence of heavy precipitation along each ray path." Or something like that... Right now, when you say, "...PRO measurements exhibit stronger signals in the presence of heavier precipitation" it is not clear what signal or measure you are referring to.

*Response:*

We agree that our original submission was insufficient in explaining the context of PRO in relation to the standard RO measurement. We thank the reviewer for pointing out this lack of background information in our introduction.

Following your suggestion, in the revised manuscript, we included a paragraph that clarifies how PRO measurement brings additional information in relation to the standard RO measurement (L41-45).

62. Slight correction: "Since RO measurements represent path-integrated quantities, a positive value of KDP is an indication of the presence of horizontally-oriented hydrometeors somewhere along the ray path".

*Response:*

Thank you for pointing our imprecise choice of words. We corrected the text as suggested (L72-74).

125. The uncertainty in (3) also relates to the natural variability in the unknown particle size distribution (PSD). Near L-band, Kdp is more related to the mass water content (3rd moment of the PSD), unlike radar reflectivity (6th moment of the PSD). Therefore, Kdp is less sensitive to variations in the PSD and the contributions from the largest drops.

For "flat enough" ice particles, Kdp can actually larger than for (similar water content of) rain media. Since the heavy liquid phase only rain is usually concentrated in a smaller domain than the rest of the (mixed or solid) phase above, a small Kdp integrated over a long path can exceed that from a larger Kdp but integrated over a shorter path. Suggest you cite Fig 11 of Turk et al 2021 to give the reader an idea of the range of values being talked about (pasted below).

*Response:*

Thank you very much for the insightful comment that helps to justify our approach.

Following your suggestion, in the revision, we acknowledge the assumption of ignoring variability of PSD in our approach (L143-145) and also included a paragraph immediately below (L149-156) which discusses why $K_{DP}$ is less sensitive to the precise specification of PSD than radar reflectivity is, and also why contributions from snow can exceed those from rain citing Figure 11 of Turk et al. (2019).

Discussion in Section 5.1 is also expanded to include more detail on GNSS geometry (L320-324).

138. Presumably the drop size in the IFS representation of LWC are so small, that these are essentially spherical anyhow (and hence contribute little to no to the net $\Phi_{DP}$ ).

*Response:*

Thank you for the suggestion. We are afraid this interpretation does not apply to IFS because, in IFS, drop size distribution is not explicitly accounted for. Our forward operator abusively uses Eq (3), which is derived for solid particles, to liquid and rain as well, assuming that the parameter $ar$ is 0.5 without using any information about droplet size.

150. Is the Geer et at formulation for converting vertical mass flux to water content specific to the IFS model, or is this a more general formulation? Readers of this manuscript may want to use this formulation with other models or reanalysis that provide vertical mass flux variables.

*Response:*

The formulation of Geer et al. (2007) is adopted in RTTOV-SCATT, which is a radiative transfer model that can be used by any NWP model not restricted to the IFS. The formulation should be applicable to any NWP model as long as vertical mass fluxes of water content are available as output. In the revision we explained this at the end of Section 2.3 (L179-181).

156. When you say, "...This configuration is essentially identical to the operational deterministic forecast", is this implying that since the simulation is done with model data so close in time to one of the operational model output time steps, that it is "essentially identical"?

*Response:*

We agree that this sentence was rather vague. What we meant here is that, first, the results can be different from the operational forecasts because of technical differences like the version of the compiler and software stack, which makes the results not identical at bit-level although there should be no significant difference at science level, and secondly, because the IFS operational suite has undergone upgrades over the period during which the examined cases happened, for some cases (actually 10 out of 11 cases) the operational forecasts have been produced from a different version of the IFS than the version used in this study. We clarified these points in the revised text (L245-248).

200. Suggest that you cite the work of Murphy et al who earlier suggested this "displacement effect" in their 2019 paper (see their discussion on observation geometry). In fact, you should cite this paper in the very beginning of this manuscript, as their work is the first study on microphysics and PRO.

M. J. Murphy, J. S. Haase, R. Padullés, S.-H. Chen, and M. A. Morris, "The Potential for Discriminating Microphysical Processes in Numerical Weather Forecasts Using Airborne Polarimetric Radio Occultations," Remote Sensing, vol. 11, no. 19, Art. no. 19, Jan. 2019, doi: 10.3390/rs11192268.

*Response:*

First of all we sincerely apologize for not being aware of the study by Murphy et al. (2019). We cited their work in the introduction (L50-53) and in Section 4.2 (L239-240) when we discuss the displacement effects.

230. I would suggest renaming this section to, "Limitations of 1D Operator". This is an important section and these limitations will magnify as model resolutions further shrink. It's important to mention that this limitation is not only valid for modeling PRO in precipitation (as you show), but in general for modeling of RO, e.g., the bending angle from the water vapor field. In areas of high pressure and fair weather (no clouds), one would expect little different between the

bending angle simulated by the 1D and 2D operators as the horizontal water vapor field is rather homogeneous. But near clouds and precipitation, where one wants to maximize the benefit of PRO or RO observations, there are also variations in the water vapor field (eg, dry/moist across frontal boundaries) such that simulated bending angles can be quite different depending upon whether the forward operator was positioned "across" or "along" the weather front.

With a coarse model grid spacing (eg, 1-degree or so), these effects may not matter too much (?). But as model resolutions get smaller and smaller, and resolve clouds and water vapor at km scales, it is important to replicate the actual ray path viewing geometry to being simulations and observations into accord.

*Response:*

Thank you for raising this important point. We completely agree that the limitation of 1D computation also applies to the regular RO observations as well, and we also agree that this will become increasingly important as the horizontal resolution of the model gets higher. In the revised text, we emphasized this point in L288-L292.

325. Referring to "horizontally sparse observations", if a dense sampling of PRO-like observations were to be available (each PRO viewing from a different relative azimuth angel), do you think that PRO would also be able to better correct for position errors?

*Response:*

Thank you for this great suggestion. We do agree that, if such observations are available, that will enable 4DVar to correct position errors, although at this point we can only speculate. In the revision we included brief discussion on such a possibility when we discuss future directions (L378-L380).

---

## Author Response (AR2)

**Response to the editor**

Reply to the review by the editor (Dr. Peter Alexander) of the revised manuscript of **AMT-2023-132: "Forward operator for polarimetric radio occultation measurements"** by Daisuke Hotta, Katrin Lonitz, and Sean Healy

We would like to express our sincerest gratitude for spending your precious time on overseeing the peer-review process of our manuscript.

Please find below our point-by-point response to you comments. As in our previous responses to the reviewers, your comments are cited in red colour, followed by our replies in black.

For convenience, we have attached a tracked-changes version of the revised manuscript (`diff.pdf`) generated with the `latexdiff` utility; in this file, newly added and deleted texts are highlighted, respectively, with blue and red colours. Line numbers and page numbers shown in the response below refer to those in the `diff.pdf` file.

We hope that our revision have addressed your concern, and thank you again for your careful assessment.

**Point-by-point response**

Referee 1 stated: "Although the agreement is still not accurate in the quantitative sense, this can be interpreted as both the uncertainty of the detailed hydrometeor field, and a margin associated with the oblateness, and probably the tilt, of hydrometeors, quantities with large uncertainty.". Some kind of argument like this one needs to be included in Section 5, but of course your perspective may be different from that idea. Or please clarify if you believe that this type of statement has already been contemplated in the discussion section.

*Response:*

We agree that we missed to address this point raised by Referee 1 in our previous revision. In the initial submission, we placed emphasis on the consistency of the simulated and observed $\Phi_{DP}$ profiles for AR cases, as shown in upper two rows of Figure 1, and then investigated why the discrepancy between the two is so large in the TC cases. However, on a close look, we do agree with Referee 1 that quantitative agreement is not as good as typical observations that are routinely assimilated in NWP systems, especially with the simulation's visible tendency to overestimate in comparison to the observations. As Referee 1 suggests, we think these quantitative mismatches can be attributed either to the imperfection of the simplistic $K_{DP}$-WC relation, especially the assumption of constant $ar$ or to the uncertainty of hydrometeor fields in the short-range forecasts of the IFS. We acknowledged the former point in Section 2.2 in the previous manuscript when we discussed our WC-to-$K_{DP}$ model, but we agree that we need to reiterate it in section 5.

In the revised manuscript, we summarised these points in section 5 (L295-299) and also slightly modified wordings in the other section to make the manuscript consistent throughout (L221-222). We also took this opportunity to proofread the entire manuscript once again and corrected a few typographic/stylistic errors (L140,L167,L390).